# Genome-Wide Identification and Analysis of *NF-Y* Gene Family Reveal Its Potential Roles in Stress-Resistance in *Chrysanthemum*

Rongqian Hu, Mengru Yin, Aiping Song [ID], Zhiyong Guan, Weimin Fang, Fadi Chen and Jiafu Jiang *[ID]

State Key Laboratory of Crop Genetics and Germplasm Enhancement, Key Laboratory of Landscaping, Ministry of Agriculture and Rural Affairs, Key Laboratory of Biology of Ornamental Plants in East China, National Forestry and Grassland Administration, College of Horticulture, Nanjing Agricultural University, Nanjing 210095, China
* Correspondence: jiangjiafu@njau.edu.cn

**Abstract:** Nuclear factor Y (NF-Y) is a class of transcription factors (TFs) with various biological functions that exist in almost all eukaryotes. In plants, heterotrimers composed of different NF-Y subunits are numerous and have different functions that can participate in the regulation of plant growth at various stages. However, *NF-Y* genes have not been systematically analyzed in chrysanthemum, only involving several NF-Y members. In this study, forty-six NF-Y members were identified from the diploid species *Chrysanthemum seticuspe*, including eight NF-YA members, twenty-one NF-YB members, and seventeen NF-YC members. These *NF-Y* genes were analyzed for their physicochemical characteristics, multiple alignments, conserved motifs, gene structure, promoter elements, and chromosomal location. Phylogenetic analysis revealed that only two gene pairs in *C. seticuspe* underwent gene duplication events. The Ka/Ks ratios were both less than one, indicating that the two pairs underwent purifying selection. Promoter element analysis showed that multiple abiotic stress and hormone response elements were present in the *CsNF-Y* genes, suggesting that these genes play an important role in the response to stress, growth, and development in plants. Further validation of candidate genes in response to drought regulation using RT-qPCR demonstrated that *CsNF-Y* genes in *C. seticuspe* play an important role in drought regulation.

**Keywords:** *Chrysanthemum*; NF-Y; gene structure; cis-element; stress



## 1. Introduction

Nuclear factor Y (NF-Y), a common transcription factor (TFs) in plants, animals, and other eukaryotes, can bind the CCAAT-box (CBF) of eukaryotic promoters and heme-activated protein (HAP) [1]. NF-Y consists of three distinct subunits: NF-YA (CBF-B/HAP), NF-YB (CBF-A/HAP3), and NF-YC (CBF-C/HAP5) [2–4]. NF-YA, NF-YB, and NF-YC activate or repress downstream gene expression by forming heterotrimeric complexes or by interacting with other regulatory factors [5–8]. In eukaryotic organisms such as yeast and mammals, where each gene encodes a subunit, NF-YB and NF-YC proteins form a heterodimer in the cytoplasm, which is then transferred to the nucleus to interact with NF-YA to form an active heterotrimeric complex [9–11]. However, in plants, after events such as whole-genome duplication (WGD), NF-Y has continued to expand in evolution, forming a highly conserved sub-unit of the same type encoded by multiple genes [12–15]. Therefore, there are numerous heterotrimers formed by plant NF-Ys, and their functions differ.

Moreover, NF-YB/NF-YC can form a heterotrimeric complex NF-CO with CONSTANS (CO) in plants, which has a CCT domain highly similar to NF-YA [12]. The CORE (CCACA) element was first identified in the *FT* promoter and recruited CO to bind the promoter of *FT* to regulate plant flowering, and it was later found to be a key binding element for the B-box family (zinc finger proteins) to regulate flowering [16,17]. Several studies have shown that the NF-Y/NF-CO complex, formed by NF-YB and NF-YC with NF-YA/CO,



regulates *FLOWERING LOCUS T* (*FT*) expression by binding to CCAAT-box or CORE elements in the upstream promoter region of *FT*, thereby regulating plant flowering [17–21]. In addition, NF-YB/NF-YC can interact with DELLAs proteins in the gibberellin signaling pathway to directly regulate the transcription of *SUPPRESSOR OF OVEREXPRESSION OF CONSTAN1* (*SOC1*) and participate in the regulation of plant flowering [22]. In Arabidopsis, ABA-response element (ABRE)-binding factors (ABF3 and ABF4) interact with NF-YCs to promote flowering by inducing *SOC1* transcription under drought conditions [23]. Subunits of NF-YB and NF-YC interact with each other in *Oryza sativa*; OsNF-YC2 and OsNF-YC4 proteins regulate flowering by interacting with OsNF-YB8/10/11 [24]. Among these, *OsGhd8* (*OsNF-YB11*) delays flowering under long-day (LD) conditions but promotes flowering under short-day (SD) conditions by regulating *OsEhd1*, *OsRFT1*, and *OsHd3a* expression [25]. It was recently found that CiNF-YA1 could specifically bind to the *CiFT* promoter by forming a complex with CiNF-YB2 and CiNF-YC2 to promote flowering of adult lemon (*Citrus limon*) under drought and low temperatures [26]. In woody plants, *RhNF-YC9* can be inhibited by ethylene to regulate petal expansion in rose (*Rosa hybrida*) [27].

Furthermore, *NF-Y* genes are involved in the regulation of many important physiological processes in plants, including responses to stress, embryo and chloroplast development, and photosynthesis. NF-YBs could interact with NF-YCs and then recruit distinct NF-YAs to form complexes that bind the CCAAT element in the *CHALCONE SYNTHASE 1* (*CHS1*) promoter and influence fruit color by regulating flavonoid biosynthesis during tomato (*Solanum lycopersicum*) fruit ripening [28]. AtNF-YCs mediate Arabidopsis photomorphogenesis by epigenetically regulating hypoblast elongation under light and through negative regulation of brassinosteroids (BRs) biosynthesis and signal transduction [29]. AtNF-YC3/4/9 interact with RGL2, a transduction repressor of GA signaling, and co-target the key gene *Abscisic acid-insensitive 5 (ABI5)* in the ABA signaling pathway to regulate Arabidopsis seed germination [30]. *LEAFY COTYLEDON 1 (LEC1)* and *LEC1-LIKE (L1L)* were identified as NF-YB members, involved in the regulation of embryogenesis and seed development in Arabidopsis [31–34]. *OsHAP3A* (*OsNF-YB2*) and its homologues *OsHAP3B* (*OsNF-YB3*) and *OsHAP3C* (*OsNF-YB4*) were shown to be required for chloroplast function [35]. Both *AtNF-YA2/3/7/10* and *OsNF-YA7* have been shown to be induced by drought stress [36,37]. In addition, *AtNF-YA 2/3/7/10* in Arabidopsis have been shown to regulate plant growth by regulating carbon metabolism-related genes, and overexpression of *NF-YAs* can retard plant senescence [36]. The heterotrimer formed by GmNF-YC14 with GmNF-YA16 and GmNF-YB2 activates the PYRABACTIN RESISTANCE 1 (GmPYR1)-mediated abscisic acid [38] signaling pathway and enhances drought tolerance in *Glycine max*, while heterologous transformation and overexpression of *GmNF-YC14* in Arabidopsis can enhance Arabidopsis drought and salt tolerance [39]. Overexpression of *StNF-YC9* could enhance drought tolerance in *Solanum tuberosum* [40]. *AtNF-YA5* is highly expressed in Arabidopsis after drought or abscisic acid [38] treatment, whereas overexpression of *AtNF-YA5* leads to reduced leaf water loss and enhanced drought tolerance [41]. Heterologous transformation of Arabidopsis with *PwNF-YB3* from *Picea wilsonii* may enhance its tolerance to salt, drought, and osmotic stress by regulating CBF-dependent pathway genes [42]. Studies in Arabidopsis, maize, chrysanthemum, poplar, rice, and other species have demonstrated that *NF-YB*s can regulate plant drought tolerance [37,43–46].

As important ornamental plants, cut flowers and groundcover chrysanthemums are widely used in production and life and occupy a very important position in the international flower trade. Because of the complex evolutionary background of *Chrysanthemum morifolium* [13], most chrysanthemum genomes are too large. The diploid species *C. seticuspe* (2n = 18) is closely related to *C. morifolium*, which provides a basis for in-depth study of *C. morifolium* [38]. However, there are few reports on the functional studies of NF-Y family members in *C. seticuspe*. Here, the physicochemical properties, multiple sequence alignment, conserved motifs, gene structure, promoter elements, and chromosomal locations of *CsNF-Y*s were systematically analyzed. The function and status of *CsNF-Y* genes in chrysanthemum growth, development, and stress resistance were further clarified.

## 2. Experimental Materials and Methods

### 2.1. Plant Material and Treatment

Seedlings were planted in a vermiculite: nutrient soil (3:1 matrix) and cultured for five weeks under long-day conditions. The seedlings were selected with the same growth and number of knots, pulled out of the soil, washed the root soil with clean water, and placed in clean water for pre-cultivation for 24–48 h to adapt to the liquid environment. For drought treatment, seedlings were placed in a solution containing 15% PEG6000 with three biological replicates per treatment.

### 2.2. Identification and Analysis of CsNF-Y Family Genes in C. seticuspe

The genome sequence and annotation data of *C. seticuspe* are available at PlantGarden "https://plantgarden.jp/en/list/t1111766/genome (accessed on 15 July 2022)". The NF-YA/B/C HMM models (PF02045 and PF00808) obtained from the Pfam database "https://pfam.xfam.org/ (accessed on 15 July 2022)" were used to identify CsNF-Y protein members. Using the NF-Y protein sequences of Arabidopsis and rice obtained from TAIR "https://www.arabidopsis.org/ (accessed on 16 July 2022)" and PlanTFDB "http://planttfdb.gao-lab.org/ (accessed on 16 July 2022)", further BLAST searches were performed to set the threshold $E < 1e^{-10}$ to obtain the CsNF-Y gene family sequences. The obtained sequences were submitted to the UniProt database "http://www.uniprot.org/ (accessed on 20 July 2022)" for de-redundancy and 46 *CsNF-Y* genes were identified. Analysis of physicochemical properties such as molecular weight (MW) and isoelectric point [27] and prediction of subcellular location were identified using Prot-Param "https://web.expasy.org/protparam/ (accessed on 29 July 2022)" and WoLF PSORT "https://wolfpsort.hgc.jp/ (accessed on 29 July 2022)", respectively.

### 2.3. Phylogenetic Analysis of CsNF-Y Family Genes

A phylogenetic tree of CsNF-Y, AtNF-Y, and OsNF-Y proteins was constructed using MEGA-X software based on the maximum likelihood method and then modified using iTOL "https://itol.embl.de/ (accessed on 5 August 2022)".

### 2.4. Multiple Sequence Alignment, Gene Structure and Conserved Motif Analysis of CsNF-Y Protein

A sequence alignment analysis of CsNF-Y protein sequences was performed using Jalview "https://issues.jalview.org/secure/Dashboard.jspa (accessed on 15 August 2022)". Conserved motifs of *CsNF-Y*s were identified using the Multiple Em for Motif Elimination (MEME) 5.0.2 online program "https://meme-suite.org/meme/doc/meme.html (accessed on 20 August 2022)". The online program Conserved Domain Database (CDD) "https://www.ncbi.nlm.nih.gov/Structure/bwrpsb/bwrpsb.cgi (accessed on 20 August 2022)" identifies the CsNF-Y conserved domains. A visualization of exon-intron positions and conserved motifs was performed using TBtools software [47].

### 2.5. Genome Chromosomal Location and Gene Duplication Parameter Analysis

*CsNF-Y*s locus information was obtained from genome annotation data. TBtools mapped the location of genes in chromosomes and analyzed gene covariance and gene replication event parameters.

### 2.6. Analysis and Statistics of Stress Response Elements of CsNF-Y Promoters

The 2kb sequences upstream of the initiation codon (ATG) of the *CsNF-Y*s were extracted using the Gtf/Gff3 sequences Extract function of TBtools. The elements in the sequence were predicted using PlantCARE "http://bioinformatics.psb.ugent.be/webtools/plantcare/html/ (accessed on 2 September 2022)" and visualized with the Basic BioSequence View function of TBtools [47].

### 2.7. RNA Extraction and RT-PCR Analysis

The total RNA was extracted from different tissues of *C. seticuspe* using a Quick RNA isolation kit (HUYUEYANG BIOTECHNOLOGY, Beijing, China). Gel electrophoresis (1.5% agar) was used to test the integrity levels of the extracted RNA. Then, the extracted RNA was employed as a template with the PrimeScript$^{TM}$RT reagent Kit with gDNA Eraser (Perfect Real Time) (Takara Biomedical Technology, Tokyo, Japan) for first-strand cDNA synthesis. A real-time quantitative PCR of *CsNF-Ys* was performed on a Light Cycler 96 System (Roche, Switzerland) using SYBR Green$^{®}$ Fast qPCR Mix (Takara, Japan). Primers were designed using Primer Premier 5.0, and the primer sequences are presented in Table S1. The PCR thermal cycling conditions were as follows: 95 °C for 2 min and 45 cycles of 95 °C for 10 s, 58 °C for 10 s, and 72 °C for 20 s. Elongation factor 1 alpha (EF1$\alpha$) [48–50] was used as the internal reference gene for data normalization. Three technical replicates of three biological replicates were used for each analysis. Finally, the relative expression of *CsNF-Y*s was calculated using the $2^{-\Delta\Delta Ct}$ method.

## 3. Results

### 3.1. Identification of NF-Y Family Genes in Chrysanthemum

The conserved models of NF-Y and protein sequences from *Arabidopsis thaliana* and *Oryza sativa* were used as queries to identify NF-Y in *C. seticuspe*. After the removal of incomplete and redundant sequences, forty-six *CsNF-Y* genes were identified, including eight *NF-YA*, 21 *NF-YB*, and seventeen *NF-YC* genes. Bioinformatics data of *CsNF-Y* genes were analyzed, including the number of amino acid residues, the theoretical molecular weight [43], and the theoretical isoelectric point [27], as shown in Table 1. The CsNF-Y proteins varied in length and physicochemical properties, the amino acid number of the shortest CsNF-Y protein is only 91aa (CsG_LG1.g57056.1), and the longest amino acid number of CsNF-Y protein reaches 1370aa (CsG_LG4.g59802.1); the molecular weights were between 10.52 kDa (CsG_LG1.g57056.1) and 154.75 kDa (CsG_LG4.g59802.1), and the pI values ranged from 4.67 (CsG_LG4.g63571.i1) to 9.78 (CsG_LG1.g35675.i1). The subcellular localization and the heat map (Supplementary Figure S1) of CsNF-Y proteins revealed that most of the subcellular localization of CsNF-Y proteins was located in the nucleus, while the others were located in the cytoplasm and chloroplast or other subcellular organelles, indicating that CsNF-Y proteins may function in the cytoplasm and nucleus to regulate plant growth and development at various stages.

**Table 1.** Properties of the predicted *NF-Y* genes in *C. seticuspe*.

| Sequence ID | PL (aa) | MW (kDa) | pI | Subcellular Localization | Arabidopsis |
|---|---|---|---|---|---|
| CsG_LG1.g19367.1 | 288 | 32.17 | 5.19 | chlo: 4, mito: 4, golg: 2 | AtNF-YC |
| CsG_LG1.g19403.1 | 288 | 32.24 | 5.2 | chlo: 4, mito: 4, golg: 2 | AtNF-YC |
| CsG_LG1.g35675.i1 | 125 | 14.39 | 9.78 | nucl: 11, extr: 2, cysk: 1 | AtNF-YA |
| CsG_LG1.g35676.i1 | 193 | 21.80 | 8.82 | nucl: 13, chlo: 1 | AtNF-YA |
| CsG_LG1.g35742.i1 | 299 | 32.57 | 5.23 | chlo: 4, nucl: 3.5, mito: 3 | AtNF-YC |
| CsG_LG1.g49936.i1 | 174 | 19.24 | 6.21 | nucl: 14 | AtNF-YB |
| CsG_LG1.g51885.1 | 310 | 35.12 | 8.47 | cyto: 7.5, cyto_nucl: 5 | AtNF-YC |
| CsG_LG1.g57056.1 | 91 | 10.52 | 9.26 | cyto: 7, chlo: 3, plas: 3 | AtNF-YC |
| CsG_LG2.g19134.1 | 151 | 16.59 | 7.66 | nucl: 4, cyto: 4, chlo: 2 | AtNF-YB |
| CsG_LG2.g19144.1 | 179 | 19.94 | 6.17 | chlo: 9, nucl: 2, cyto: 1 | AtNF-YB |
| CsG_LG2.g19160.1 | 179 | 19.92 | 5.8 | chlo: 10, nucl: 1, cyto: 1 | AtNF-YB |
| CsG_LG2.g20363.1 | 263 | 29.42 | 5.34 | nucl: 9, chlo: 2, cyto: 2 | AtNF-YC |
| CsG_LG2.g27837.1 | 198 | 22.53 | 5.76 | nucl: 14 | AtNF-YB |
| CsG_LG2.g40057.i1 | 290 | 32.16 | 6.72 | nucl: 14 | AtNF-YA |
| CsG_LG2.g50775.i1 | 164 | 17.71 | 6.08 | nucl: 12, pero: 2 | AtNF-YB |
| CsG_LG4.g21245.1 | 160 | 17.80 | 7.54 | nucl: 13, pero: 1 | AtNF-YB |
| CsG_LG4.g27765.1 | 289 | 32.54 | 6.25 | cyto: 8, nucl: 4, chlo: 1 | AtNF-YC |

**Table 1.** *Cont.*

| Sequence ID | PL (aa) | MW (kDa) | pI | Subcellular Localization | Arabidopsis |
|---|---|---|---|---|---|
| CsG_LG4.g59802.1 | 1370 | 154.75 | 8.03 | chlo: 4, vacu: 3, plas: 2 | AtNF-YB |
| CsG_LG4.g63571.i1 | 155 | 17.20 | 4.67 | cyto: 6, mito: 4, chlo: 3 | AtNF-YB |
| CsG_LG5.g13905.1 | 235 | 26.50 | 7.02 | nucl: 11, cyto: 1, plas: 1 | AtNF-YC |
| CsG_LG5.g23964.1 | 228 | 25.86 | 6.15 | nucl: 6, extr: 3, cyto: 2 | AtNF-YC |
| CsG_LG5.g24551.1 | 280 | 31.32 | 9.12 | nucl: 9.5, cyto_nucl: 6 | AtNF-YA |
| CsG_LG5.g30681.1 | 380 | 42.64 | 6.52 | cyto: 14 | AtNF-YC |
| CsG_LG5.g42217.1 | 123 | 13.82 | 5.4 | cyto: 6, nucl: 2.5, chlo: 2 | AtNF-YB |
| CsG_LG5.g51977.1 | 209 | 21.95 | 6.31 | nucl: 14 | AtNF-YB |
| CsG_LG5.g58078.1 | 258 | 28.87 | 6.96 | nucl: 6, cyto: 5, chlo: 1 | AtNF-YC |
| CsG_LG5.g62437.1 | 442 | 49.77 | 5.91 | cyto: 10, chlo: 4 | AtNF-YC |
| CsG_LG6.g14713.1 | 136 | 15.09 | 7.74 | mito: 9, chlo: 3, nucl: 2 | AtNF-YB |
| CsG_LG6.g44069.1 | 157 | 17.69 | 4.93 | chlo: 6, nucl: 5, extr: 2 | AtNF-YB |
| CsG_LG6.g69448.1 | 199 | 21.67 | 8.36 | nucl: 14 | AtNF-YB |
| CsG_LG7.g07056.1 | 195 | 21.99 | 5.64 | nucl: 6, mito: 4, cyto: 3 | AtNF-YB |
| CsG_LG7.g66779.i1 | 195 | 20.73 | 5.95 | nucl: 14 | AtNF-YB |
| CsG_LG7.g73142.1 | 213 | 23.85 | 6.17 | cyto: 7, nucl: 2, plas: 2 | AtNF-YB |
| CsG_LG8.g04800.1 | 264 | 29.60 | 6.56 | nucl: 13, plas: 1 | AtNF-YA |
| CsG_LG8.g08763.i1 | 325 | 35.09 | 9.26 | nucl: 14 | AtNF-YA |
| CsG_LG8.g09275.1 | 297 | 32.99 | 9.2 | nucl: 14 | AtNF-YA |
| CsG_LG8.g12663.1 | 134 | 15.36 | 9.45 | nucl: 13, chlo: 1 | AtNF-YC |
| CsG_LG8.g27812.1 | 157 | 17.15 | 5.76 | nucl: 8, mito: 6 | AtNF-YB |
| CsG_LG8.g38215.1 | 236 | 25.67 | 9.02 | nucl: 14 | AtNF-YB |
| CsG_LG8.g62859.1 | 219 | 23.91 | 5.14 | nucl: 9, chlo: 2, cyto: 2 | AtNF-YC |
| CsG_LG8.g67227.1 | 205 | 21.70 | 6.3 | nucl: 14 | AtNF-YB |
| CsG_LG9.g35177.1 | 432 | 48.67 | 8.94 | vacu: 5, golg: 3, chlo: 2 | AtNF-YC |
| CsG_LG9.g39194.1 | 229 | 25.97 | 5.88 | nucl: 7, cyto: 2, plas: 1.5 | AtNF-YC |
| CsG_LG9.g51759.i1 | 300 | 33.32 | 9.19 | nucl: 13, chlo: 1 | AtNF-YA |
| CsG_LG9.g52368.i1 | 299 | 32.83 | 5.25 | nucl: 3.5, cyto_nucl: 3.5 | AtNF-YC |
| CsG_LG9.g60513.1 | 160 | 18.15 | 5.76 | nucl: 8, cyto: 3, chlo: 1 | AtNF-YB |

The abbreviated letters PL in the second column of the table refer to protein length. The values after subcellular localization in column 4 represent the number of times the homologous subcellular localization of the gene has been predicted in different species. The abbreviated letters nucl means nuclear; cyto means cytoplasmic; chlo means chloroplast; mito means mitochondrial; vacu means vacuolar; pero means peroxisomal; extr means extracellular; plas means plasma membrane; golg means golgi apparatus.

### 3.2. Phylogenetic Analysis of CsNF-Y Proteins

To further investigate the phylogenetic relationships of NF-Ys among *C. seticuspe*, *A. thaliana*, and *O. sativa*, a phylogenetic tree of 46 CsNF-Y, 34 OsNF-Y, and 36 AtNF-Y protein sequences was constructed using the maximum likelihood method with MEGA-X software, and iTOL was used for further modification, as shown in Figure 1. From the results of the phylogenetic tree, we found that all NF-Y proteins could be naturally clustered into three clades (NF-YA, NF-YB, and NF-YC). Ten AtNF-YAs and eleven OsNF-YAs belonged to the NF-YA subunit (blue-filled); eight CsNF-Ys, eleven OsNF-YBs, and thirteen AtNF-YBs belonged to the NF-YB subfamily with twenty-one CsNF-Ys (orange-filled); and seventeen CsNF-Ys were related to thirteen AtNF-YCs and twelve OsNF-YCs and were considered to belong to the NF-YC subunit (green-filled). Notably, CsNF-Y members within the NF-YA and NF-YB subunits are closely related to AtNF-Ys and OsNF-Ys, respectively. In contrast, CsNF-YCs in the NF-YC subunit were more closely related to AtNF-YCs, with 10 CsNF-YC members being highly homologous to AtNF-YC9/3. Four CsNF-YCs have evolutionary homology with AtNF-YC11, and two CsNF-YCs are closely related to AtNF-YC12 and AtNF-YC13.

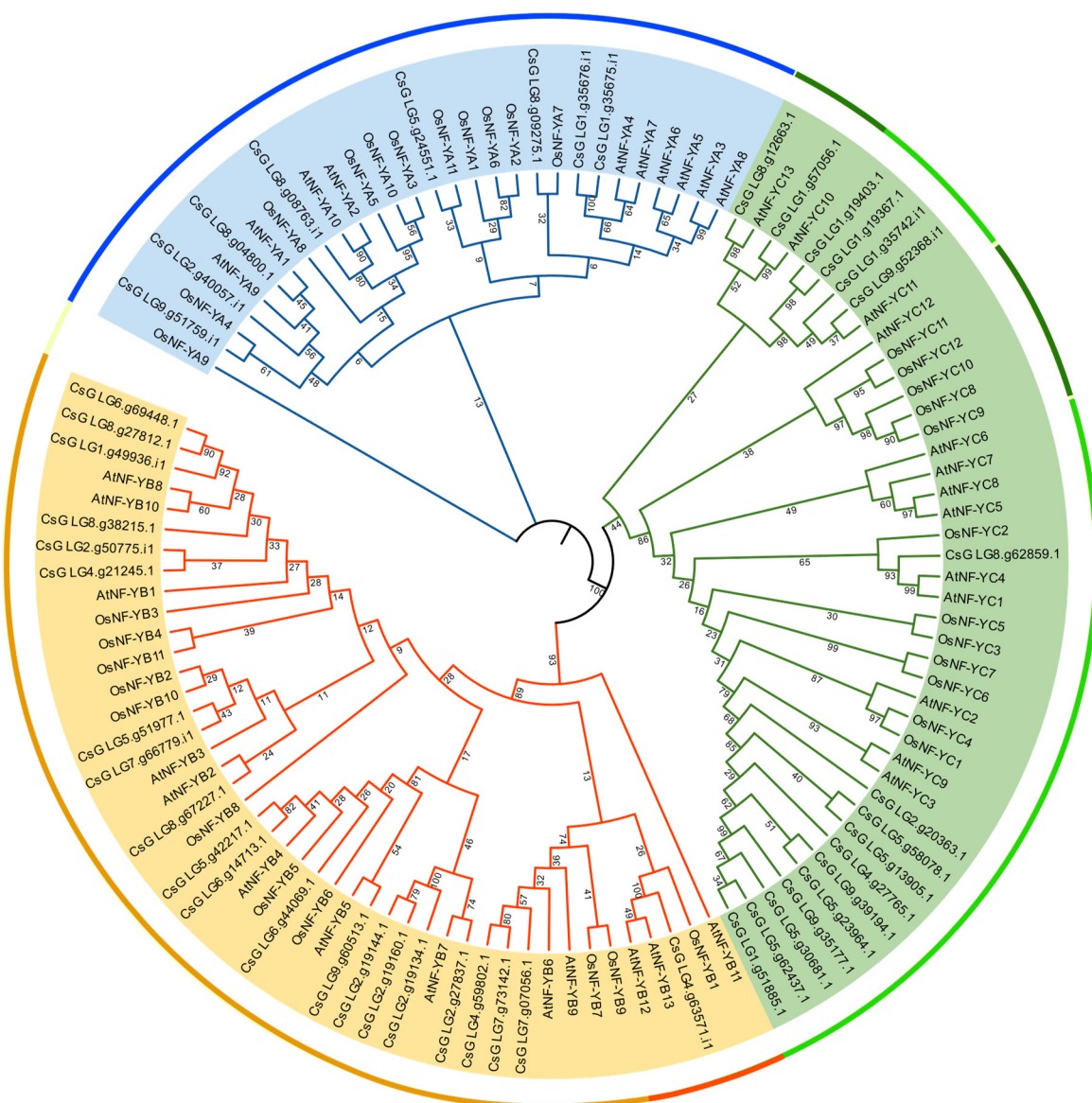

**Figure 1.** Phylogenetic analysis of NF-Y protein from *C. seticuspe*, *A. thaliana*, *O. sativa*. Full-length protein sequences of NF-Ys were used to construct the tree by using MEGA-X based on the maximum-likelihood (ML) method. Subfamily are highlighted with different colors. The blue represents NF-YA subfamily, the orange represents NF-YB subfamily, and the green represents NF-YC subfamily.

### 3.3. Multiple Sequence Alignment of CsNF-Y Genes in Chrysanthemum

*CsNF-Y* genes were divided into three major subunits: NF-YA, NF-YB, and NF-YC. Multiple sequence alignments were analyzed using MEGA-X and displayed in JalView. As shown in Figure 2, all sequences harbored conserved core regions [4,51]. Most of the CsNF-YA-conserved regions were approximately fifty-two amino acids and consists of two α-helices. The first domain, A1, mediates the NF-YB/NF-YC interaction [52]. The second, A2, is the DNA-binding domain, which includes three histidine (H) and three arginine (R) residues that are essential and absolutely conserved, which can recognize the CCAAT-box on DNA [11,53]. The conserved regions of the CsNF-YB and CsNF-YC subfamilies were approximately eighty-five and seventy-five amino acids in length, respectively, which contain four α-helices. The CsNF-YB conserved regions contained the NF-YA interaction region, DNA-binding domain, and NF-YC interaction region (Figure 2b). In previous studies, NF-YB subunits were classified into two types based on the similarity of conserved domain sequences: the LEC1 type, consisting of LEC1 and

LEC1-LIKE (L1L), and the non-LEC1 type, including the rest of the subunits [32]. The analysis showed that *C. seticuspe* possesses three LEC1-type CsNF-YB subunit members. Specifically, the LEC1-type conserved structural domain possesses 15 shared residues that are different from the conserved residues at equivalent positions in the non-LEC1-type conserved structural domain, as shown in Figure 2. The conserved domain of CsNF-YCs contained the NF-YA interaction structure, DNA-binding domain, and NF-YB interaction structure (Figure 2c).

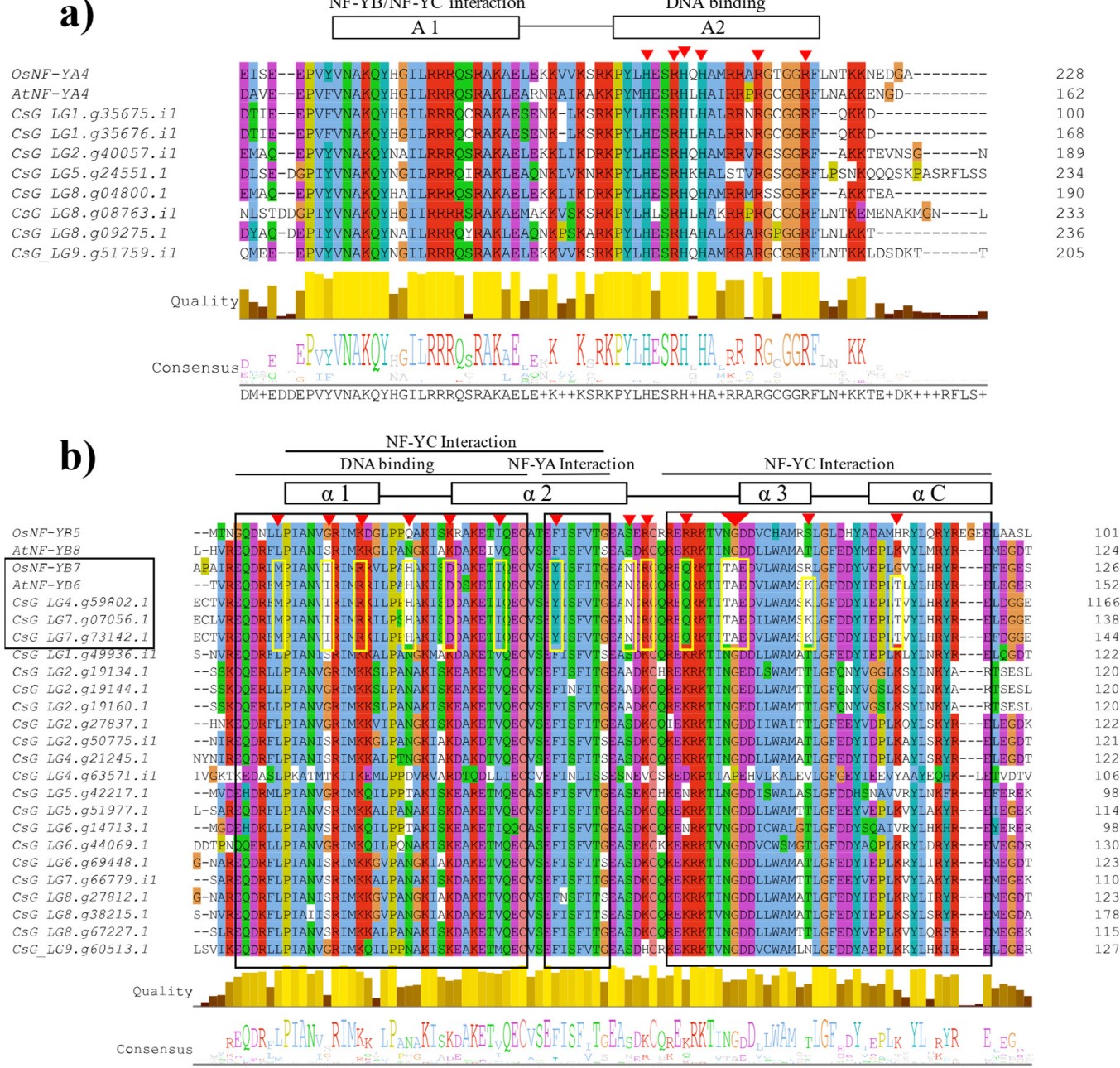

**Figure 2.** *Cont.*

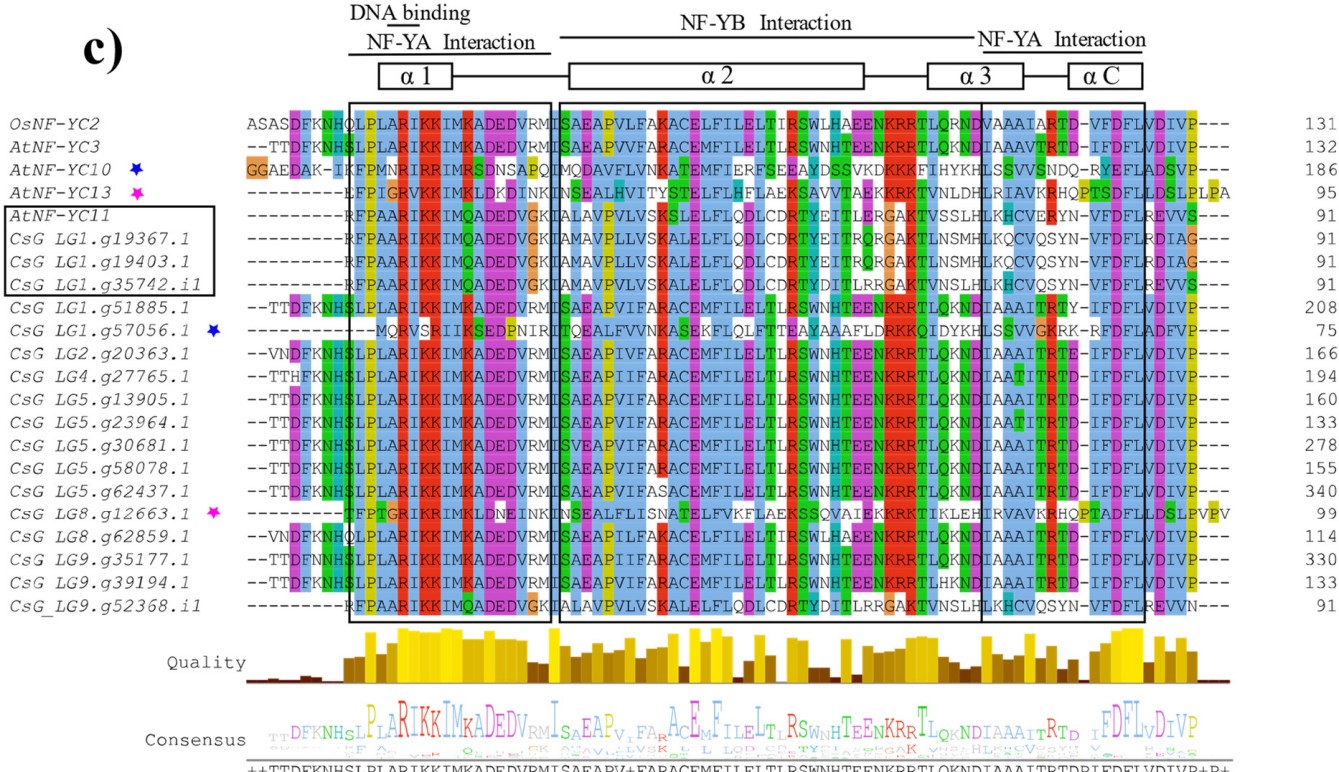

**Figure 2.** The details of conserved domains in the three subfamilies of CsNF-Ys Subunit interaction and DNA binding regions of conserved domains are marked. The numbers followed the gene number are the conserved domain ranges. Below the sequences are the amino acid enrichment analysis and above the sequences are the domain analysis. (**a**) CsNF-YA subunit; (**b**) CsNF-YB subunit; (**c**) CsNF-YC subunit: stars represent the presence of *CsNF-YC* genes with structural similarity to the particular branch of AtNF-YC and NF-YCs with structurally similar stars of the same color.

### 3.4. Analysis of Gene Structure and Conserved Motifs of CsNF-Y Genes in Chrysanthemum

The 46 full-length amino acid sequences of CsNF-Y were used to construct a phylogenetic tree using the MEGA-X software. Forty-six CsNF-Ys were divided into three subunits based on phylogenetic analysis of *C. seticuspe*, *O. sativa*, and *A. thaliana* (Figure 3a). Gene structure analysis can provide insights into the evolution of gene families; therefore, the distribution of exons and introns in CDS was analyzed using TBtools. Green squares and black lines represent the exon and intron regions, respectively (Figure 3c). The results showed that all *CsNF-YA* genes were separated by introns, and each *CsNF-YA* contained at least two introns. A few *CsNF-YBs* and *CsNF-YCs* did not harbor introns, including nine *CsNF-YBs* and three *CsNF-YCs*; the remaining *CsNF-YBs* and *CsNF-YCs* contained one to eight introns. To study the diversification of the three CsNF-Y subfamilies, putative motifs of CsNF-Y proteins were analyzed using the online MEME program. The details of the ten motifs are shown in Supplementary Figure 2. The results revealed that the three CsNF-Y subfamilies had unique motif distributions (Figure 3b). Each *CsNF-YA* contained fixed motifs 3 and 5. Twenty of the twenty-one *CsNF-YBs* had structures with motifs 2, 1, and 3 in tandem, whereas only one (CsG_LG4.g63571.i1) lacked motif 3. One member of the *CsNF-YB* (CsG-LG4. g59802.1) had a strange length, and after motif and NCBI-conserved domain analysis by MEME, it was found that it had a characteristic motif structure and conserved domain of the *CsNF-YB* family, as well as other protein structural domains, presumably because the gene had been subjected to gene integration during the evolutionary process. The motifs of *CsNF-YC* were insufficiently conserved, and motifs 4 and 6–10 were only present in *CsNF-YC*. The conserved domain positions of the CsNF-Y protein sequences were analyzed and counted using NCBI for a total of 10 conserved domains

(Supplementary Figure S1). In general, each *CsNF-YA* was composed of two conserved motifs, all but one *CsNF-YB* was composed of three conserved motifs, and *CsNF-YC* contained one to seven motifs. Therefore, these results indicate that *CsNF-YA*s have more conserved gene structures and motifs than *CsNF-YB*s and *CsNF-YC*s do.

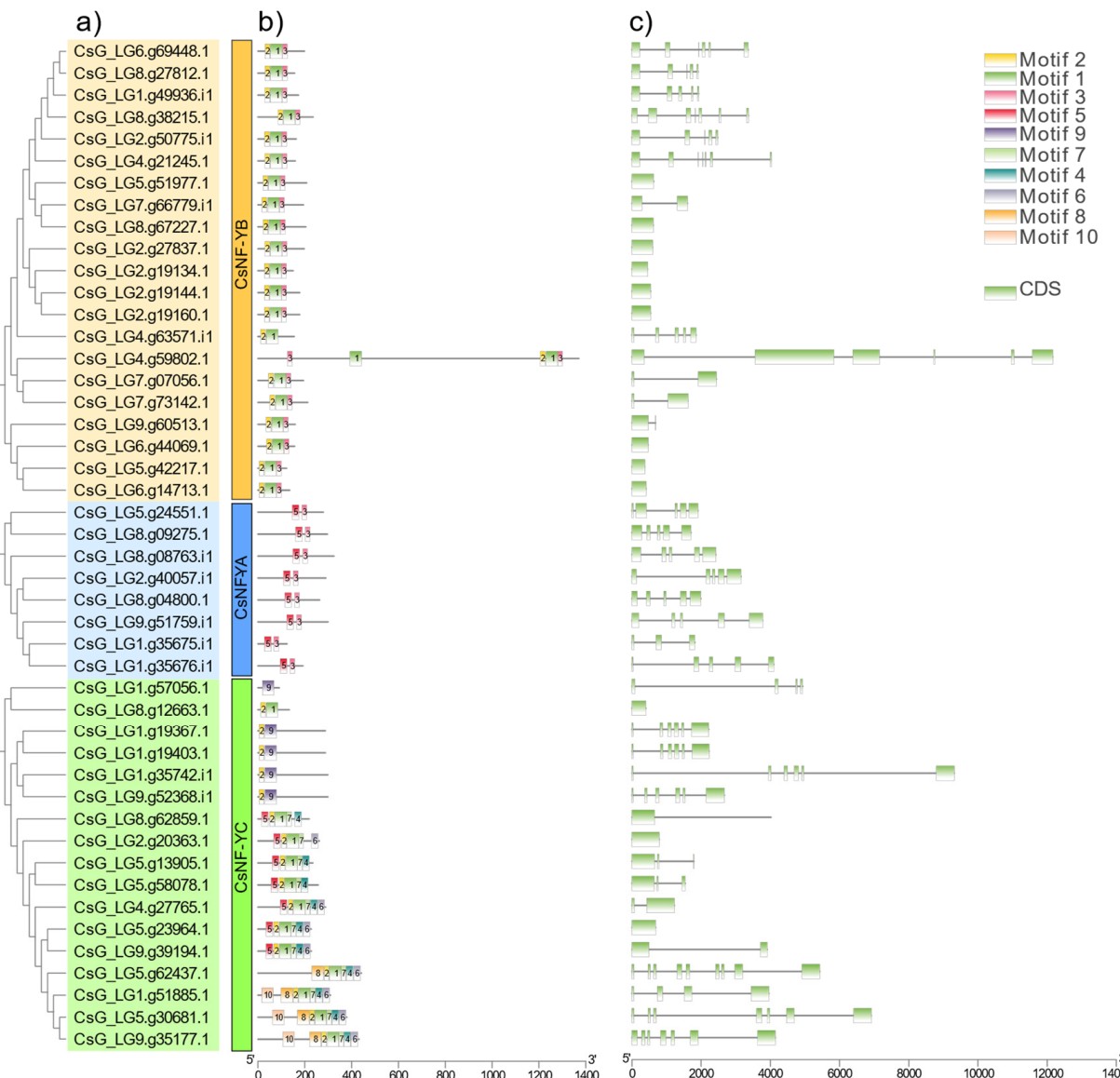

**Figure 3.** Motif structure, conserved domains and gene structure of CsNF-Y family were analyzed using MEME, NCBI and TBtools. (**a**) CsNF-Y gene was constructed using MEGA-X to build an unrooted phylogenetic tree. (**b**) Schematic diagram of motif structure in the *CsNF-Y* gene family using MEME. Different colors and numbers represent ten different motifs. (**c**) *CsNF-Y* gene structure analysis, including intron and exon distribution.

*3.5. Analyses of Chromosomal Distribution and Synteny of CsNF-Y Genes in Chrysanthemum*

All *CsNF-Y* genes were unevenly distributed in the chromosomes of *C. seticuspe* (Figure 4). *CsNF-Y* was widely distributed in CsG_LG1, CsG_LG5, and CsG_LG8, with eight members. This was followed by CsG_LG2, with seven *CsNF-Y* members. Five *CsNF-Y*s were found in CsG_LG9, and four *CsNF-Y*s were found in CsG_LG4. Three *CsNF-Y*s were located in each of CsG_LG6 and CsG_LG7. Notably, CsG_LG3 did not contain CsNF-Y.

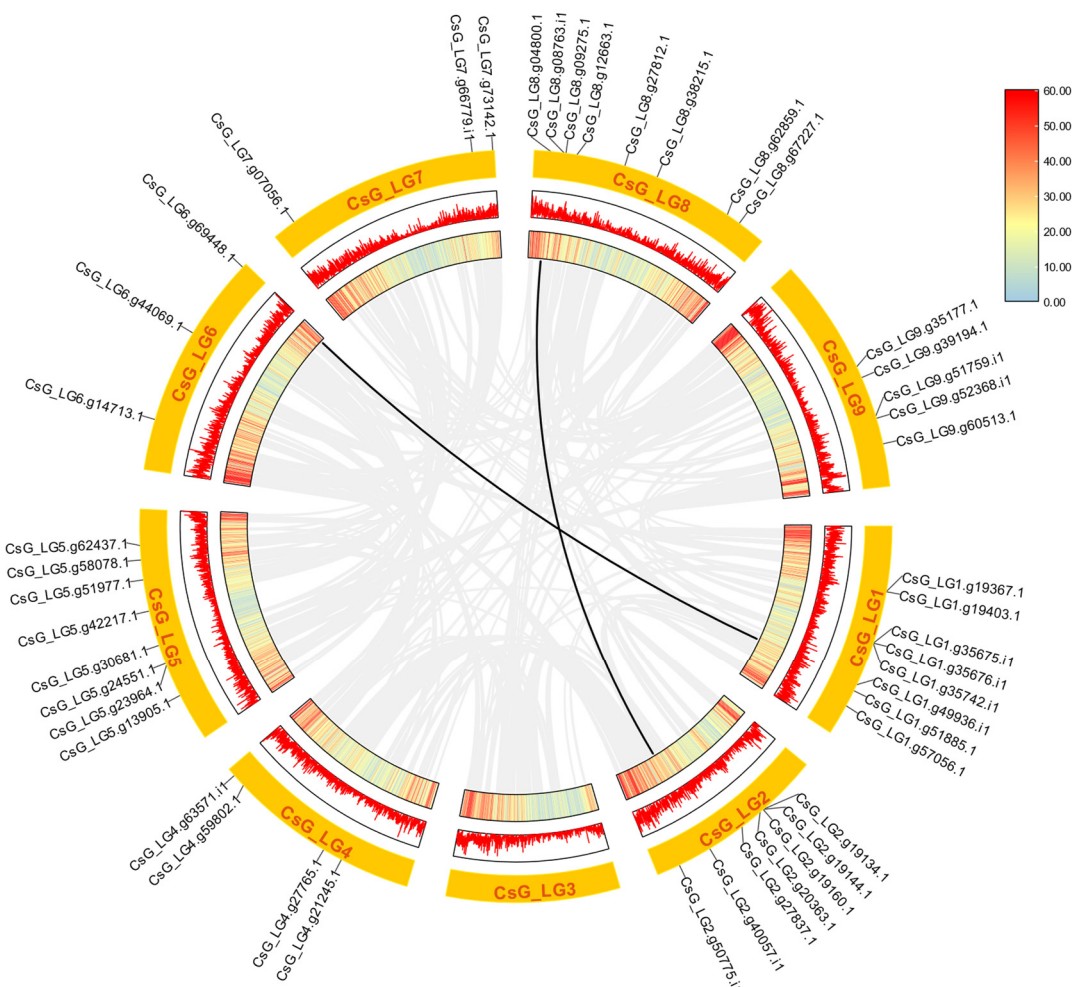

**Figure 4.** Genomic location and duplication pairs of the *CsNF-Y* gene on the chromosome of *C. seticuspe*. The yellow bands of CsG_LG1-9 represent nine chromosomes, respectively. Grey lines in the background represent collinear blocks of *C. seticuspe*; black lines highlight syntenic *CsNF-Y* gene pairs.

Tandem repeats, segment repeats, and whole-genome duplications play key roles in gene family member expansion and new function realization but are also major drivers of organismal evolution [54,55]. Collinearity analysis showed that there were two *CsNF-Y* gene pairs in *C. seticuspe*—that is, two pairs of genes underwent gene duplication events. No gene duplication event was formed by genes on the same chromosome, but two gene duplication events were formed by genes in different chromosomes, occurring in the CsNF-YA and CsNF-YB subunits.

The substitution rates of non-synonymous (Ka) and synonymous [56] are the basis for evaluating the positive selection pressure of repeated events; Ka/Ks < 1 indicates purifying selection, Ka/Ks = 1 signifies neutral selection, and Ka/Ks > 1 indicates positive selection [57]. To understand the driving forces behind the evolution of the *CsNF-Y* gene family, the parameters (Ks, Ka, and Ka/Ks ratio) of *CsNF-Y* homologous gene pairs were calculated using TBtools [47]. We found that the Ka/Ks ratios of *CsNF-Y* homologous gene pairs for both gene duplication events were much smaller than 1, proving that *CsNF-Ys* in *C. seticuspe* were selected by purification (Table S1).

### 3.6. Analysis of Cis-Elements in the Promoter Sequences of NF-Y Genes in Chrysanthemum

Multiple sequence alignment, conserved motifs, and gene structure analysis can help us understand the conserved and structural features of *CsNF-Y* family genes horizontally. Promoter elements allow us to understand the roles played by genes in plant physiological

processes. To elucidate the possible regulatory mechanisms of *CsNF-Y* genes in response to various stresses, we identified putative cis-elements in the 2kb promoter region upstream of the translation initiation site (ATG) of *CsNF-Ys*. The *CsNF-Y* promoter sequences were analyzed using the online PlantCARE program [58]. All cis-elements in the promoter regions of the *CsNF-Y* genes are shown in Figure 5. The promoters of 30 *CsNF-Y* genes contained cis-elements related to auxin, abscisic acid, ethylene, and salicylic acid. The gibberellin response elements included the GARE-motif, P-box, TATC-box, and 23 of the 46 *CsNF-Y* genes involved in the gibberellin response. The promoters of the 33 *CsNF-Ys* harbored the TGACG-motif and CGTCA-motif, which are MeJA-responsive elements. A total of forty-two and seven *CsNF-Y* gene promoters were involved in anaerobic and metabolic reactions, respectively. Nineteen *CsNF-Y* gene promoters contained low-temperature response elements. The cis-elements responding to drought include not only DRE [59], but also cis-elements ABRE, MBS (involved in the drought-induced MYB binding site), and W Box, which are also involved in drought-induced signaling and regulation of downstream gene expression [3,37]. Forty-five of the forty-six *CsNF-Y* genes contained one or more drought-responsive cis-elements in their promoter regions. Among the five *CsNF-Y* genes, one *CsNF-YA* (CsG_LG8.g08763.i1), two *CsNF-YBs* (CsG_LG2.g27837.1, CsG_LG9.g60513.1), and two *CsNF-YCs* (CsG_LG1.g35742.i1, CsG_LG5.g58078.1) were involved in circadian rhythm regulation. According to elemental statistics, almost all *CsNF-Y* genes can be involved in the regulation of plant drought resistance, and the cis-elements of anaerobic induction were abundant in the promoters of *CsNF-Ys*. In addition, a large number of *CsNF-Ys* participate in the response to the hormones ETH, SA, MeJA, and ABA. In conclusion, *CsNF-Ys* play an essential role in stress response and various stages of growth and development in plants.

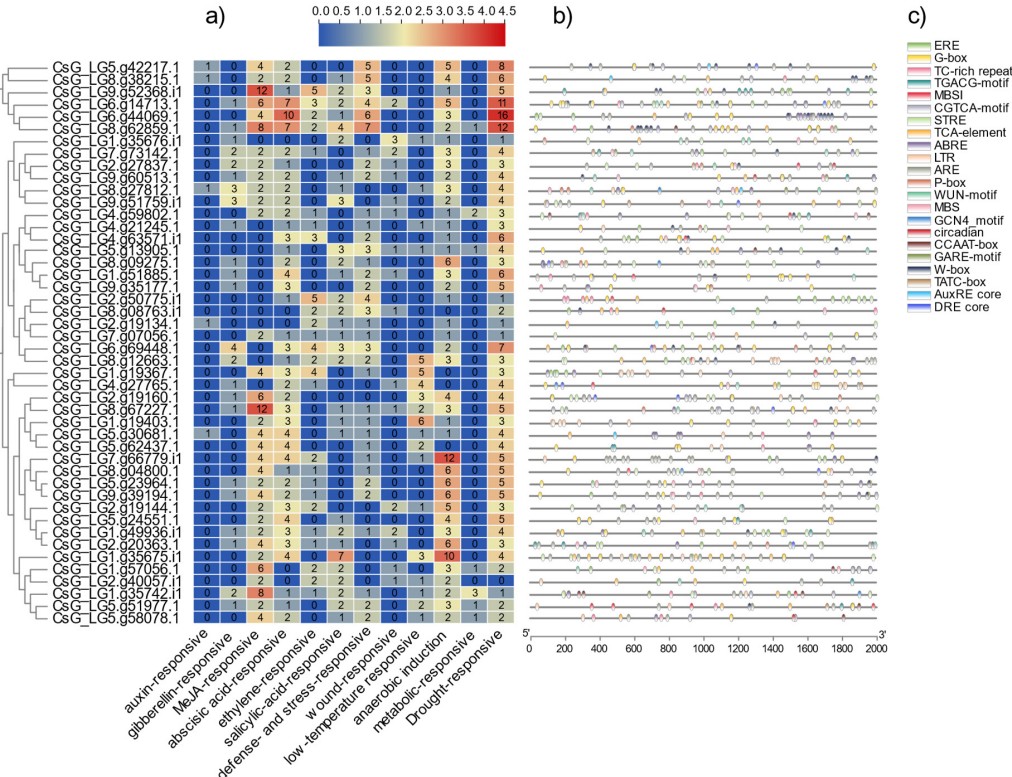

**Figure 5.** Putative cis-elements and transcription factor binding sites in the promoter regions of *NF-Y* genes from *C. seticuspe*. (**a**) The color and number of the grid indicate numbers of different cis-acting elements in these *CsNF-Y* genes. (**b,c**) The colored blocks represent different types of cis-acting elements and their locations in each *CsNF-Y* gene.

### 3.7. Expression Levels of CsNF-Y Genes in Chrysanthemum under Osmotic-Stress

Many studies have shown that *NF-Y*s are involved in drought regulation and the regulation of plant flowering under drought conditions [23,26,37,41–46,60]. Moreover, the results of the promoter analysis based on *CsNF-Y*s showed that *CsNF-Y*s play an important role in drought regulation. Therefore, further exploration of the potential functions of *CsNF-Ys* involved in plant drought regulation is necessary. Based on transcriptome data obtained under accession number PRJNA82048835 [48], a total of 39 *CsNF-Y* genes were identified and their expression pattern in different plant organs (roots, stems, leaves, ray florets and disc florets) were analysed (Figure S3). The graph showed that three *CsNF-Y* genes were constitutively expression in chrysanthemum (CsG_LG1.g35676.i1, CsG_LG1.g35742.i1, CsG_LG8.g67227.1), some *CsNF-Y*s showed high expression in floral organs during reproductive growth period, such as CsG_LG5.g13905.1, CsG_LG1.g57056.1, CsG_LG5.g58078.1, CsG_LG9.g39194.1, CsG_LG5.g23964.1 and CsG_LG1.g19403.1, two *CsNF-Y*s (CsG_LG8.g09275.1 and CsG_LG8.g04800.1) showed specifically high expression in roots, while one particular *CsNF-Y* (CsG_LG8.g62859.1) was expressed in all organs except roots.

Based on the expression pattern and promoter elements of *CsNF-Ys*, and the reports of drought-related of *CsNF-Ys* homologs [23,37,41,60], 19 candidate genes were further selected for expression level analysis after osmotic-stress treatment. We performed RT-qPCR analysis on mature third leaves of *C. seticuspe* under simulated drought stress (15% PEG 6000). Among all candidate genes, the expression levels of CsG_LG1.g35676.i1, CsG_LG1.g49936.i1, and CsG_LG8.g62859.1, were repressed by osmotic-stress, and the expression of CsG_LG8.g08763.i1 and CsG_LG1.g57056.1 were briefly repressed after osmotic-stress treatment and then gradually restored their expression levels (Figure 6). The remaining *CsNF-Y*s genes were all up-regulated by using 15% PEG treatment, among which CsG_LG1.g19403.1, was most obviously up-regulated after treatment (Figure 6). RT-qPCR showed that *CsNF-Y*s were involved in plant responses to osmotic-stress through different regulatory approaches. Taken together, the response of *CsNF-Y*s to osmotic-stress implies that these genes may play important roles in the modulation of drought-stress.

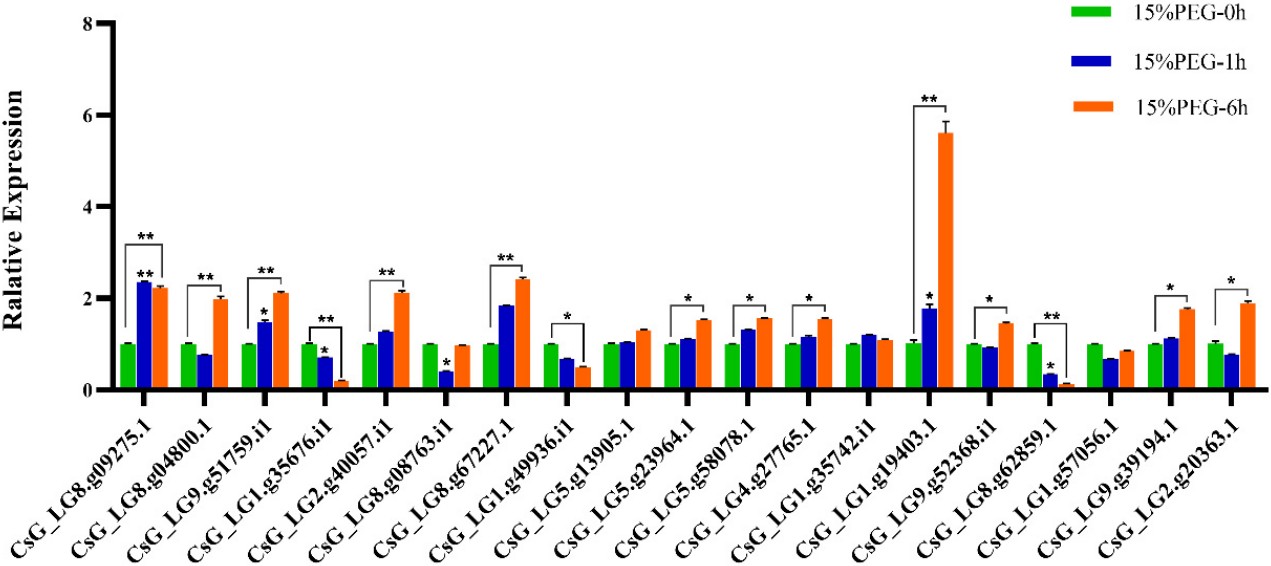

**Figure 6.** Expression patterns of *CsNF-Y* family genes in response to drought stress. The treatment concentrations were 15% PEG 6000. Values are mean $\pm$ S.D. * $p \leq 0.05$, ** $p \leq 0.01$, Sidak's test.

### 4. Discussion

Regarding to the *AtNF-Y* family genes *AtNF-YB11/12/13* and *AtNF-YC10/12/13* in *A. thaliana*, due to the inappropriate structures, previous reports have suggested that the six *AtNF-Y*s should separate from the NF-Y family [61]. The phylogenetic tree constructed using

CsNF-Ys, OsNF-Ys, and AtN-Ys showed that AtNF-YB11/12/13 and AtNF-YC10/12/13 had a distant evolutionary relationship with other members, and three CsNF-Ys (CsG_LG4.g63571.i1, CsG_LG1.g57056.1, and CsG_ LG8.g12663.1) were also found in phylogenetic trees that were closely related to AtNF-YB13, AtNF-YC10, and AtNF-YC13, respectively. No homologues of AtNF-YB11 and AtNF-YC12 were found in the C. seticuspe genome, whereas OsNF-YC8/9/10/11/12 in rice was homologous to AtNF-YC12 in Arabidopsis. The conserved structural domains of CsNF-Ys were analyzed: CsG_LG1.g57056.1 and CsG_LG8.g12663.i1 are structurally similar to AtNF-YC10/13, respectively. Four CsNF-YC members had close affinity to AtNF-YC11, including CsG_LG9.g52368.i1, CsG_LG1.g35742.i1, CsG_LG1.g19367.1, and CsG_LG1.g19403.1, but the conserved regions of these four genes are not fully consistent with the remaining CsNF-YCs. The protein sequence of AtNF-YC11 was detected by uploading it to NCBI and found to possess a conserved domain (BUR6), so CsG_LG9.g52367.i1, CsG_LG1.g35742.i1, CsG_LG1.g19367.1, and CsG_LG1.g19403.1 differed in individual amino acids in the structural domain. CsG_LG8.g62859.1 is evolutionarily homologous to AtNF-YC4. AtNF-YC3/9 has multiple homologous genes in C. seticuspe. It has been reported that AtNF-YC3/4/9 interacts with AtNF-YB2/3 and CO to form a complex involved in CONSTANS(CO)-mediated, photoperiod-dependent flowering in Arabidopsis [62]. AtNF-YC3/4/9 also interacts with AtABF3/4 to induce AtSOC1 expression to promote plant flowering [23]. CsG_LG2.g40057.i1 and CsG_LG8.g04800.1 are homologous to AtNF-YA9 and AtNF-YA1, respectively. In Arabidopsis, AtNF-YA1, 5, 6, and 9 are functionally redundantly involved in the regulation of male gametophyte development, embryogenesis, seed development, and post-germination growth [63]. OsNF-YA7 is a homologue of CsG_LG8.g09275.1, and its expression was induced by drought rather than ABA, and overexpression of OsNF-YA7 enhanced drought tolerance in rice [37]. In addition, NF-YA genes are extensively involved in root and leaf development and flowering regulation [26,64–66]. The NF-YB subunits in the LEC type genes LEC1 (AtNF-YB9, AT1G21970; OsNF-YB7, LOC_Os02g49370) and L1L (AtNF-YB6, AT5G47670), which serve as key genes for embryonic development [32–34,67,68], contain three structurally similar homologues in C. seticuspe (CsG LG4.g59802.1, CsG LG7.g73142.1, CsG LG7.g07056.1). CsG LG5.g51977.1, CsG LG7.g66779.i1, and CsG_LG8.g67227.1 belong to a class of genes homologous to AtNF-YB2 and AtNF-YB3. Overexpression of AtNF-YB2 and AtNF-YB3 enhances drought and heat tolerance in Arabidopsis [60]. AtNF-YB2 and AtNF-YB3 induce FT expression during long days to promote flowering [69]. In addition, it was demonstrated that AtNF-YB2 and AtNF-YB3 are highly conserved in flowering function with the homologue OsNF-YB8/10/11 in rice [70].

We found that all CsNF-YA genes were separated by introns, and each CsNF-YA contained at least two introns. A few CsNF-YBs and CsNF-YCs did not harbor introns. CsNF-YAs have more introns than CsNF-YBs and CsNF-YCs. Introns play multiple functions in eukaryotic genomes and may act as mutational buffers to protect coding sequences from randomly occurring deleterious mutations [71]. Therefore, it is speculated that the structure of the NF-YA subunit gene is more stable. Cis-elements are closely related to gene function and regulatory mechanisms. Analysis of promoter cis-elements can help to quickly understand the roles of genes in plant physiological and biochemical regulatory networks, which is beneficial to gene research. The elements within the 2 kb promoter range before ATG of 46 CsNF-Y genes are shown in Figure 5, among which G-box, ABRE, ARE, and ERE were abundant in the promoters of CsNF-Ys. Studies have shown that ABRE, an ABA response element, is involved in the regulation of drought and salt stress [72]. There were reports claimed that ERE was an ethylene-responsive cis-element [73], and ARE was an anaerobic induction. Multiple stress-responsive elements have been predicted in the CsNF-Y promoters, indicating that CsNF-Y genes play a crucial role in plant stress responses.

The NF-Y gene family, which are crucial regulators of plant growth and physiology, had been carefully and critically analyzed in Arabidopsis, rice, soybean, peach, maize, alfalfa, buckwheat, Populus (as a woody model plant), potato, tomato, grape, barley, citrus, and so

on [8,51,74–84]. In *Arabidopsis*, *AtNF-Y* had been declared to be involved in seed germination, embryo development, hypocotyl elongation, flowering, drought stress, salt stress, heat stress, and other pathways to regulate plant growth [18,19,22,23,29,30,32,34,36,41,85]. In rice, *OsNF-Y* genes had been implicated in the regulation of seed development, salt tolerance, drought tolerance, and disease resistance [37,67,86–89]. Maize and soybean are extremely important cash crops for yield improvement and resistance to environmental stress [44,80]. Previous studies have expounded on the important properties of plants, such as drought tolerance, from many aspects. However, there are few reports on *NF-Y* family genes in chrysanthemum, a highly ornamental horticultural crop. Wang et al. [45] investigated *CmNF-YB8* in polyploid chrysanthemum (*Chrysanthemum morifolium*) with respect to drought tolerance and revealed that *CmNF-YB8* mainly altered stomatal movement and cuticle thickness in the leaf epidermis, thereby affecting drought resistance. The function of other *NF-Y* genes in chrysanthemum, particularly the role of *NF-Y* in stress resistance, deserves further attention.

Drought, the second most costly weather event in the world, is a natural disaster with the greatest impact on animals and plants. It is worthwhile to study how plants can escape drought and complete their life cycle through their own regulatory mechanisms when drought occurs. We performed RT-qPCR on candidate *CsNF-Y* genes in *C. seticuspe* treated with 15% PEG, and the results showed that *CsNF-Y* genes exhibited different response patterns. Genes with close affinity to *AtNF-YC3/9* (CsG_LG2.g20363.1, CsG_LG5.g58078.1, CsG_LG5.g13905.1, CsG_LG4.g27765.1, CsG_LG9.g39194.1, and CsG_LG5.g23964.1) showed significant upregulation of expression after drought treatment. CSG_LG8.g62859.1, the homologous gene of *AtNF-YC4*, was consistently repressed. *AtNF-YC3/4/9* have been reported to be involved in the drought-responsive redundant regulation of Arabidopsis growth [23]; therefore, whether the same functional redundancy exists in chrysanthemum deserves further investigation. Among the many *CsNF-Y* genes regulated by drought, CsG_LG1.g35676.i1 and CsG_LG1.g49936.i1 expression was also suppressed by drought. CsG_LG1.g35676.i1 and CsG_LG1.g49936.i1 are closely associated with *AtNF-YA4/7* and *AtNF-YB8/10*, respectively. The functions of *AtNF-YA4/7* and *AtNF-YB8/10* in drought have not yet been reported. Therefore, it is important to clarify the function of CsG_LG1.g35676.i1 and CsG_LG1.g49936.i1. In chrysanthemum, *CmNF-YB8* was reported to be repressed by drought, and overexpression of *CmNF-YB8* impaired drought resistance [45]. CsG_LG8.g09275.1 expression was upregulated by drought induction, and the homologous gene *OsNF-YA7* in rice was reported to be upregulated by drought induction to mediate drought resistance in an ABA-independent manner [37]. *OsNF-YA4* is induced by ABA, but its expression is not affected by drought [37]. While CsG_LG9.g51759.i1, a homologue of *OsNF-YA4,* was drought-induced in *C. seticuspe*, its expression was significantly upregulated. The role of CsG_LG9.g51759.i1 in chrysanthemum requires further investigation. In conclusion, *CsNF-Ys* play an essential role in the physiological regulation of stress in *C. seticuspe*.

## 5. Conclusions

In this study, we identified 46 *CsNF-Y* genes in the *C. seticuspe* genome, including eight *CsNF-YA*, 21 *CsNF-YB*, and 17 *CsNF-YC* genes. The physicochemical properties, multiple alignments, phylogenetic relationships, gene structure, conserved motifs, promoter elements, and distribution on chromosomes were analyzed. Multiple alignments revealed that the CsNF-YB subunit contains three LEC-type genes with 15 shared residues in the conserved structural domains. Synthetic analysis identified two pairs of duplicated genes in the *C. seticuspe* genome, demonstrating that whole-genome duplication events occurred in the *CsNF-Y* gene family. The analysis showed that the promoters of *CsNF-Y* genes contain a large number of hormone and stress response elements. In addition, analysis of expression pattern showed that some of the *CsNF-Y* genes were expressed with tissue specificity. *CsNF-Y* genes in *C. seticuspe* showed different response patterns under 15 % PEG treatment, the response of *CsNF-Ys* to osmotic-stress implies that these genes may

play important roles in the modulation of drought-stress. The results of this study provide valuable reference information for further studies on the functions of *CsNF-Y* genes.

**Supplementary Materials:** The following supporting information can be downloaded at: https:// www.mdpi.com/article/10.3390/horticulturae9010070/s1, Figure S1: Heat map of predicted subcellular location of *CsNF-Ys*; Figure S2: Details of 10 Motifs of CsNF-Y family genes; Figure S3: Expression patterns of CsNF-Y genes in five chrysanthemum organs; Table S1: Ka/Ks ratios of tandemly and segmentally duplicated *CsNF-Ys*; Table S2: Primers sequences used in this study.

**Author Contributions:** Conceptualization, R.H. and J.J.; Investigation, R.H. and M.Y.; Supervision, A.S. and J.J.; Writing—original draft, R.H.; Writing—review and editing, Z.G., W.F., F.C. and J.J. All authors have read and agreed to the published version of the manuscript.

**Funding:** This research was funded by the Project of Seed Breeding Revitalization in Jiangsu province (JBGS [2021]034) and the Project of Agricultural Science and Technology Independent Innovation in Jiangsu province (CX (20)1001).

**Institutional Review Board Statement:** Not applicable.

**Informed Consent Statement:** Not applicable.

**Data Availability Statement:** The datasets generated for this study are available on request to the corresponding author.

**Acknowledgments:** We would like to thank Makoto Kusaba, Hiroshima University, for sharing the material of *Chrysanthemum seticuspe* Gojo-0.

**Conflicts of Interest:** The authors declare no conflict of interest.

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
