# Peer review of "Genome-Wide Identification and Analysis of NF-Y Gene Family Reveal Its Potential Roles in Stress-Resistance in Chrysanthemum"

_horticulturae, doi:10.3390/horticulturae9010070_

Round 1

Reviewer 1 Report

This is a very good, straightforward, but coherent descriptive biology - with a little bit of molecular data. Perfectly fine. 

The presentation is well done, professional, to the point. 

see notes, including corrections + suggestions in the PDF directly.

QUIBBLE: I disagree with "drought" being a PEG, osmotic treatment. I think that is overselling, and this is not necessary - the authors should change to "mimic drought using a high osmotic PEG treatment" and then change it in the text to something like 'osmotic-stress treatment" or just 15% PEG. There are many other changes with either treatment causes that are not captured by the other.

Abbreviations here and there were off. see notes.

gene vs PROTEIN were sometimes inverted. notes were made.

referencing of the methods, section 2.6, was a bit poor - I think everything came from TB tools - but that was not very clear. see notes.

I note that the iTOL tree and alignment could be shared online forever, free of charge - if the authors choose to do so. see notes in iTOL - but that is an option, definitely not a must! I am not affiliated. 

Reviewer 2 Report

Reviewer’s comments

Overall, the manuscript is clear and pleasant to read. The experiments and bioinformatics analysis are well described.

Overall, it seems to me that the references cited in the introduction are relatively old. Could you add more recent references?

In details, the points which have to be revised:

Line 44:

The abbreviation FT has to be explained.

Can you detail the “CORE elements”?

Line 46-48:

Please, indicate also the full name of the proteins the first time you cite them.

Line 50:

Oryza sativa instead of O. sativa

Line 52:

The abbreviations LD and SD have to be explained.

Line 55:

Lemon instead of citrus?

Perhaps the Latin name of the lemon should be given because initials are used in gene names.

Lines 68-69:

Not clear for me. The same AtNF-YA genes are cited twice. The link between the 2 sentences (“In contrast”) seems not adapted. Is it really the same set of genes?

Line 72:

The abbreviation PYR1 has to be explained.

Line 73:

Can you precise Gycine max for soybean to understand the abbreviation Gm in the gene name?

Line 78:

You must precise the name of the plant for which the abbreviation is Pw.

Line 145:

For the RT-qPCR experiments, did you test other references genes along with EF1alpha?

Line 151:

I do not understand the word “modes” in the context of the sentence.

Table 1:

The abbreviation PL should be explained. Protein Length, I suppose?

Figure 1: Lines 186 and 187:

Represents instead of represent (3 times).

Lines 189 and after:

3.3. Multiple Sequence Alignment of CsNF-Y Genes in Chrysanthemum. 189

Adding representatives of Arabidopsis and rice NF-Y genes to the multiple alignments would likely be of interest to assess the conservation of amino acid residues between species?

Lines 204-205:

“15 shared residues”: this is not clearly visible on the figure 2. Perhaps something is lacking  in the legend?

Lines 206-207:

“The conserved domain of CsNF-YCs contained an NF-YA interaction structure” : 2 NF-YA interaction domains can be see in the figure ?

Figure 2:

What means the symbol alpha (a) in the figure? This is not explained.

In the legend of the figure, domains instead of domians (2 times)

Figure 3:

In the legend, domains instead of domain.

Line 268:

Table S1 instead of Figure S1.

Line 282:

Figure 5 instead of Figure 6.

Line 293:

“Among the five CsNF-Y genes”: I cannot understand “five” as I cannot see from which genes you are talking. Can you give the name of the genes?

Line 312 and Figure 6 (line 322):

You say that you performed expression analysis of 46 genes but the results are only shown for 19 of them. If so, can you add that the expression levels of the other genes were stable in your study?

Line 335:

“CsG_LG1.g57056.1 and CsG_LG8.g12663.i1 are structurally similar to AtNF-YC10/13”:

This is not shown in the figures of the article and moreover no reference is given on this subject.

Line 341:

Amino acid instead of bases?

Lines 343-345:

The conclusions that the author would like to suggest are not clearly stated.

Lines 350-351:

“The CsG_LG8.g09275.1 homolog OsNF-YA7 enhances drought tolerance in rice in an ABA-independent manner”

This is not well formulated. And not right.

Line 352:

“The NF-YB subunits in the LEC type genes LEC1 and LEL1, which serve as key genes for embryonic development”:

The names of the homologous NF-YB genes of Arabidopsis and rice (their numbers) should be given.

Line 353:

LEL1? Do you mean LEC1-LIKE (or L1L)?
